

# Spatiotemporal variations of atmospheric black carbon concentration and its correlation with meteorological and environmental factors in Xinjiang, China, from 2010 to 2022

Xiang Xie[1,2,3,*], Zhujun Zhao[4,5,*], Shuting Li[2,3], Zhongqi Lu[6], Lijuan Chen[1] and Jiangjiang Cai[2,3]

[1] Aletai Meteorological Bureau, Aletai, China

[2] Field Scientific Experiment Base of Akdala Atmospheric Background, China Meteorological Administration, Urumqi, China

[3] Institute of Desert Meteorology, China Meteorological Administration, Urumqi, China

[4] Shihezi Meteorological Bureau of Xinjiang, Shihezi, China

[5] Wulanwusu Special Test Field base of National Integrated Meteorological Observation/Wulanwusu Ecology and Agrometeorology Observation and Research Station of Xinjiang/Wulanwusu Agrometeorological Experiment Station, Shawan, Xinjiang, China

[6] School of Chemistry and Chemical Engineering, Changji University, Changji, China

[*] These authors contributed equally to this work.

Corresponding author
Zhujun Zhao,
putaojing1995@163.com

## ABSTRACT

Black carbon (BC) is a pollutant produced by the combustion of fossil fuels and biomass fuels, which has a huge impact on regional climate, atmospheric environment, and human health. In this study, based on MERRA-2 reanalysis data and ground-based observation data, the Mann-Kendall (MK) test and random forest (RF) model were used to explore the spatiotemporal variation characteristics of atmospheric BC concentration in Xinjiang, China and its correlation with meteorological and environmental covariates in 2010–2022. The results showed that the use of MERRA-2 reanalysis data to explore the spatiotemporal variation characteristics of BC concentration in Xinjiang had high reliability (relative average deviation (RAD) = 0.65). From 2010 to 2022, the annual average concentration of atmospheric BC in Xinjiang was $195.40 \pm 15.55$ ng/m$^3$, and the multi-year average change rate was $-0.05\%$. The winter season had the highest atmospheric BC concentration ($145.52 \pm 39.31$ ng/m$^3$), followed by autumn ($124.95 \pm 28.82$ ng/m$^3$), spring ($74.05 \pm 9.96$ ng/m$^3$), and summer ($73.41 \pm 5.69$ ng/m$^3$). The atmospheric BC concentration had a significant spatial variation, showing two high-BC-concentration areas on the northern slope of the Tianshan Mountains (centered on Urumqi-Changji-Shihezi region) and the urban agglomeration around the Tarim Basin (centered on Kashgar). The RF model analysis showed that meteorological factors including snow depth, surface temperature, and humidity as well as environmental factors including $NO_2$, $PM_{10}$, and $SO_2$ were the main factors affecting the BC concentration. This work is of great significance for clarifying the accumulation and spatial distribution characteristics of atmospheric BC in northwest China and the factors influencing the atmospheric BC concentration, and helps to raise

public attention to the increasingly serious climate change and public health problems caused by BC.

## INTRODUCTION

Air pollution is getting worse due to the rapid development of the economy and the use of large amounts of fossil fuels (*Shen et al., 2017*). As an important component of atmospheric aerosols, black carbon (BC) aerosols come from multiple sources. The natural sources are mainly forest fires, volcanic eruptions, *etc.*, and the anthropogenic sources are use of fossil fuels, biomass combustion in agricultural production, automobile exhaust, *etc.* These are the key factors influencing atmospheric BC concentration (*Qin, Tang & Wen, 2001*). Black carbon is chemically stable, and it can stay in the atmosphere for about 4–12 days. Besides, it also has a porous structure with a large specific surface area that makes it easy to adsorb other atmospheric pollutants (heavy metals, aromatic hydrocarbons, *etc.*) to form compound pollutants. The accumulation of these pollutants seriously impacts the human respiratory system, causing cardiovascular diseases and even cancer (*WHO, 2012*; *Galdos et al., 2013*). In addition, BC has a strong ability to absorb visible light and some infrared light, which affects the albedo of ice, snow, and clouds, and causes positive radiative forcing and exacerbate global warming (*Wu et al., 2009*). BC particles can also reduce atmospheric visibility, and participate in photochemical reactions and gas-particle transformation in the atmosphere (*Zhang et al., 2021*; *He et al., 2022*). Therefore, BC has a serious impact on regional atmosphere and human health.

The spatiotemporal variation characteristics of atmospheric BC and its correlation with meteorological and environmental factors have attracted much attention (*Schuster et al., 2005*; *Shindell & Faluvegi, 2009*; *Zhang et al., 2020*). Currently, the atmospheric BC concentration is mainly obtained by ground-based monitoring, numerical simulation, remote sensing, and reanalysis data. The ground-based monitoring relies on optical absorption and other methods to analyze the carbon content in the sample and obtain the BC concentration. Numerical simulation uses the atmospheric chemical transport model, combined with emission sources and transport conditions, to quantitatively describe the change of air pollutants. Satellite monitoring employs remote sensing technology to obtain the aerosol optical thickness (AOD) to inverse the pollutant concentrations. There are pros and cons to these approaches. To increase the accuracy of atmospheric BC concentration observations, in 2017, NASA's Global Modeling and Assimilation Office (GMAO) released the second edition of the Modern-Era Retrospective Analysis for Research and Applications (MEAAR-2), a long-term atmospheric reanalysis data containing atmospheric BC concentrations, which has been widely used in various studies. For example, *Zhuravleva et al. (2020)* used the near-surface BC concentration data of MERRA-2 for climate change assessment in northern Russia, and found that

the assessment error was about 30%, and that the aerosol transmission could cause an abnormal increase in BC concentrations (*Vinogradova et al., 2021*). Therefore, the MERRA-2 BC concentration data can be used to better understand the spatiotemporal variation characteristics on global-, national-, and regional-scale atmospheric BC pollution. The BC concentration shows significant spatial and temporal variations. For example, on the global scale, the atmospheric BC concentration in the northern hemisphere is higher than that in the southern hemisphere (*Cao, 2022*). Besides, BC concentrations are higher in the mid-latitudes due to large pollutant emissions, especially in cities and industrial areas in Europe and East Asia (*Zhang, Ma & Zheng, 2008*). On the national scale, the atmospheric BC concentration in eastern China is higher than that in the western China (*Zhang, Ma & Zheng, 2008*). The Beijing-Tianjin-Hebei region and Henan province have the highest annual average BC concentration and the fastest annual growth rate from 2000 to 2016 according to a study based on MERRA-2 BC data (*Xu et al., 2020*). Atmospheric BC concentration is not only affected by anthropogenic emissions, but also closely related to topographic and meteorological factors (*Ye, Ma & Ha, 2018*). High wind speeds, low temperatures, and low relative humidity may promote the dispersion of pollutants (*Wei & Wang, 2018*; *Zhang et al., 2020*), and the atmospheric BC concentrations in regions with frequent human activities are higher than those in regions with less human activities (*Cheng et al., 2018*).

Xinjiang Uygur Autonomous Region, located in northwest China, is a core area for implementing the Belt and Road Initiative. The main landform types include deserts, Gobi, plateaus, mountains, oases, and urban agglomerations. In recent years, the rapid economic development in Xinjiang has triggered a rapid increase in the consumption of fossil fuels, although there has also been a strong promotion of clean energy use. The changes of BC aerosols can reflect the contribution of anthropogenic activities to air pollution. In this study, the spatiotemporal variation characteristics of atmospheric BC concentration in Xinjiang in 2010–2022 were explored using MERRA-2 reanalysis data and ground-based observation data. Furthermore, based on meteorological and environmental observation data, random forest (RF) model was used to analyze the correlation between atmospheric BC concentration and meteorological and environmental factors, to identify the main factors causing BC concentration changes. This study will inform the formulation of regional environmental policies to improve air quality and achieve sustainable development, deepen the understanding of the factors influencing atmospheric BC concentrations, and raise public awareness of the increasingly serious climate change and public health problems caused by BC.

## MATERIALS AND METHODS

### MERRA-2 reanalysis data

The MERRA-2 dataset assimilates satellite aerosol (AOD) datasets based on the Goddard Earth Observing System (GEOS)-5 model and its data assimilation system (ADAS−5.12.4). The AOD datasets include the AOD dataset of the moderate resolution imaging spectrometer (MODIS), the corrected AOD dataset of the advanced very high resolution
radiometer (AVHRR), the AOD dataset of the multi-angle imaging spectroradiometer (MISR), and the AOD dataset of the aerosol robotic network (AERONET). In addition, the MERRA-2 dataset also assimilates several remote sensing-retrieved datasets and the observation datasets from sondes and aircraft.

The MERRA-2 dataset includes 72 vertical layers from the surface to about 80 km and a variety of datasets, including clouds, radiation, hydrological cycles, and ozone, which is useful for comprehensive analysis of land, air, and sea. The spatial distribution data of atmospheric BC concentrations since 1980 (spatial resolution: $0.5° \times 0.625°$; format: NetCDF) could be extracted from the MERRA-2 dataset downloaded from NASA's Earth Science Data website (https://science.nasa.gov/earth/data/). The data format conversion, cropping, and regional statistics were completed in ArcGIS 10.8, to obtain the average atmospheric BC concentrations in Xinjiang from 2010 to 2022.

### Ground-based data

In this study, the accuracy of the atmospheric BC concentrations in Xinjiang of the MERRA-2 dataset was verified by using the ground-based observation data (October 1, 2010–December 31, 2022) provided by the Akdala Atmospheric Background Station (47°10′N, 87°58′E, 562 m a.s.l., Fig. 1). The sampling instruments (AE-31 and AE-33 aethalometers) were installed on the roof of a building of the Akdala Station. The inlet was 3 m away from the ground. Because the light absorption by BC accounts for 90%–95% of the total light absorption by aerosols, the aethalometers use the absorption characteristics of BC to measure the mass concentration of BC (*Wang et al., 2012*). The aethalometers could obtain BC concentration data of 7 wavelengths (370, 470, 520, 590, 660, 880, and 950 nm). The BC concentration measured at 880 nm was selected as the observed concentration (minute-level data), and then the monthly average BC concentrations were calculated. Since one grid of MERRA-2 BC data was enough to cover the observation station, the monthly average data of a MERRA-2 grid close to the observation station were selected for comparison, to verify the applicability and accuracy of the MERRA-2 BC data. The correlation was strong if the correlation coefficient (R) was in the range of 0.6–0.8, moderate if the R was in the range of 0.4–0.6, and weak if the R was in the range of 0–0.4.

### Meteorological and environmental data

In this study, the meteorological data and environmental pollutant concentration data (January 1–December 31, 2022) provided by the Akdala Atmospheric Background Station were used to explore the correlation between atmospheric BC concentration and meteorological and environmental factors. The daily meteorological data included wind speed (m/s), land surface temperature (°C), humidity (%), snow depth (cm), sunshine hours (h), *etc.*, and the daily environmental pollutant concentration data included $PM_{10}$ (ug/m$^3$), $PM_{2.5}$ (ug/m$^3$), $NO_2$ (ppb), $SO_2$ (ppb), CO (ppb), and $O_3$ (ppb) concentrations.

### Research methodology
#### *Random forest model*

Random forest is a classifier formed by many decision trees. The data in the raw set are randomly selected to form a new sample set through bootstrap resampling and a model

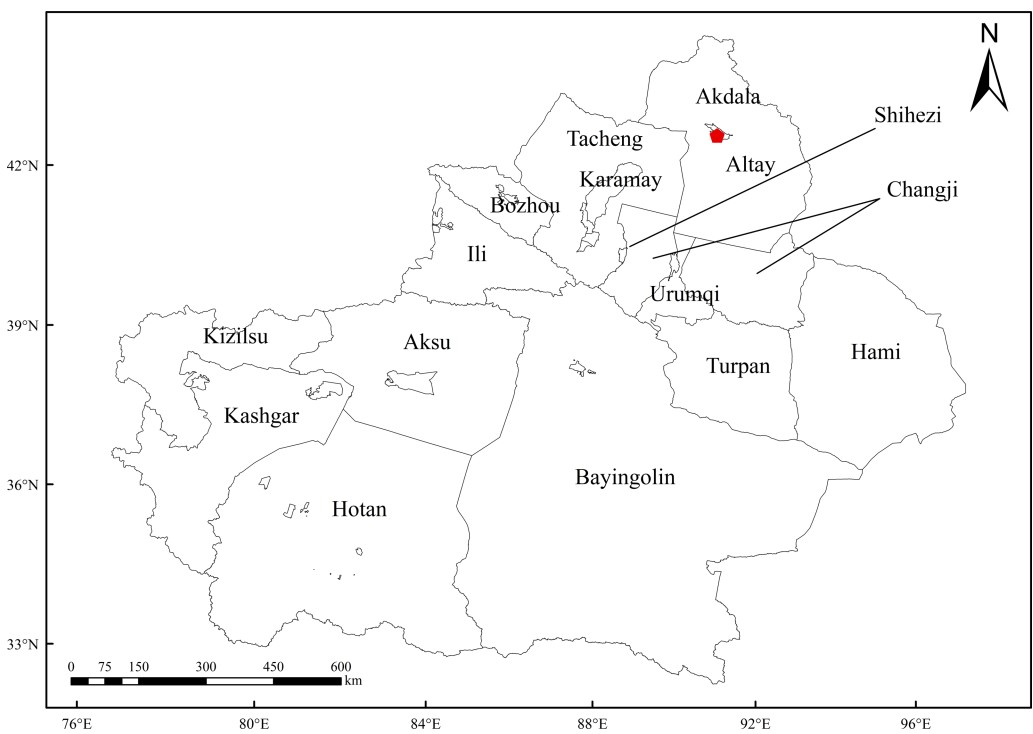

**Figure 1 Location of Akdala station.** Note: The boundary data are from China's geospatial data cloud public data.

composed of N classification trees (*Wang, Zhang & Draxler, 2009*). The stochastic method is used to determine the number of features based on the estimation error, classification ability, and correlation coefficient. It provides the relative importance of each variable, and performs importance ranking for multiple variables, which reduces balance errors and avoids overfitting (*Ruckstuhl et al., 2012*). In this study, the atmospheric BC concentration was used as the dependent variable, and the meteorological (land surface temperature, humidity, sunshine hours, snow depth, and wind speed) and environmental ($PM_{10}$, $PM_{2.5}$, $NO_2$, $SO_2$, CO, and $O_3$ concentrations) factors were used as the independent variables.

### *Evaluation of the applicability of MERRA-2 BC dataset*

Since the data for a grid in the MERRA-2 BC data represents the concentration of BC in the corresponding area, the BC concentration data in a MERRA-2 grid corresponding to the area where the observation station was located were selected for comparison, to evaluate the applicability of the MERRA-2 BC data. The regression method was used to obtain the correlation coefficient (R), root mean square error (RMSE), mean absolute error (MAE), and relative average deviation (RAD). Besides, the accuracy evaluation results were compared with those of MERRA-2 BC data used in other regions in previous studies, to determine the applicability of MERRA-2 BC data in Xinjiang. If the RAD $\geq$ 0.5, it indicates high confidence of the MERRA-2 BC reanalysis data (*Westervelt et al., 2016*). The formula for calculating each parameter is as follows:

**Table 1 Evaluation of the applicability of MERRA-2 atmospheric black carbon (BC) data to Xinjiang, China.**

| Region | MERRA-2 average BC concentration ($\mu$g/m$^3$) | BC concentration provided by the Akdala station ($\mu$g/m$^3$) | Root mean square error (RMSE) ($\mu$g/m$^3$) | Mean absolute error (MAE) ($\mu$g/m$^3$) | RAD | Correlation coefficient (R) |
|---|---|---|---|---|---|---|
| China | 3.31 | 4.07 | 2.54 | 1.71 | 0.83 | 0.61 ($p < 0.01$) |
| Northwest China | 1.28 | 2.42 | 2.54 | 1.32 | 0.53 | 0.77 ($p < 0.01$) |
| Akdala station in Xinjiang | 0.194 | 0.327 | 0.176 | 0.133 | 0.65 | 0.49 ($p < 0.01$) |
| Southwest China | – | – | 1.33 | 1.04 | 0.88 | 0.91 ($p < 0.01$) |
| Beijing | – | – | – | – | – | 0.778 ($p < 0.01$) |
| Tianjin | – | – | – | – | – | 0.72 ($p < 0.01$) |

$$RMSE = \sqrt{\frac{1}{n}\sum_{i=1}^{n}(BC_{(MERRA-2)i} - BC_{(ground-2)i})2} \tag{1}$$

$$MAE = \frac{1}{n}\sum_{n}^{1}|BC_{(MERRA-2)i} - BC_{(ground-2)i}| \tag{2}$$

$$RAD = \frac{BC_{(MERRA-2)i}}{BC_{(ground-2)i}} \tag{3}$$

where $n$ represents the total number of samples, and $i$ represents the $i$th sample.

# RESULTS AND DISCUSSION

## Evaluation of the applicability of MERRA-2 BC data in Xinjiang, China

The linear regression of the observed BC concentration at the Akdala Station and the BC concentration at the corresponding MERRA-2 grid ($N = 131$) showed that $R = 0.49$ ($p < 0.05$), the RMSE and MAE were smaller than those of China and northwest China, and the RAD was greater than that of northwest China (*Cao, 2022*) (Table 1). Therefore, the spatiotemporal variation characteristics of BC concentration in Xinjiang can be explored using the MERRA-2 reanalysis data for it has high reliability (RAD = 0.65). However, due to the impact of the spatial resolution of the reanalysis data, the observed BC concentrations were higher than the MERRA-2 data. Besides, because the Akdala Station is the atmospheric background station for northern Xinjiang and one of the regional background stations of the Global Atmosphere Watch (GAW) network of the World Meteorological Organisation (WMO), the BC concentrations are much lower than the average value in other regions (*Cao, 2022*).

## Variation of atmospheric BC concentration from 2010 to 2022

The annual average BC concentration in Xinjiang was 195.40 $\pm$ 15.55 ng/m$^3$, and the multi-year average change rate was $-0.05\%$ (Fig. 2). The annual average BC concentration was the highest (231.99 ng/m$^3$) in 2012 and the lowest (174.50 ng/m$^3$) in 2022. The

variation range of the annual average BC concentration in Xinjiang was 174.50–231.99 ng/m$^3$. The annual average BC concentration in Xinjiang is lower than the averages of western (0.38 µg/m$^3$) and eastern (2.36 µg/m$^3$) China from 1980 to 2020, and also significantly lower than that of the Beijing-Tianjin-Hebei region, Sichuan Basin, and northern Henan (average BC concentration from 1980 to 2020: 3.35–5.01 µg/m$^3$) (*Cao, 2022*). The Beijing-Tianjin-Hebei region and northern Henan have developed industry and large populations. The changes in BC concentration are closely related to human and industrial activities. The Beijing-Tianjin-Hebei region and northern Henan are developed and have a large population, while Xinjiang has a vast area, sparse population, and low GDP, so the BC concentration in Xinjiang is low (*Tiwari et al., 2013*; *Mao et al., 2016*). The BC concentration in Xinjiang showed a decreasing trend in the past 13 years, indicating an improvement in regional BC pollution, but there was an increase in BC concentration in some regions of Xinjiang. Specifically, the BC concentration in Xinjiang showed a significant increasing trend from 2010 to 2012, especially, from 2011 to 2012, the BC concentration increased by 17.90%. From 2013 to 2015, there was a rapid downward trend, with a cumulative decline of 22.52%. After 2016, there was a fluctuating trend. Anthropogenic emissions are key factors affecting atmospheric BC concentrations. For example, Xinjiang's energy industry output continued to increase from 2010 to 2012. Xinjiang's GDP in 2012 reached 750.531 billion yuan, increasing by 40.0% compared with that in 2010. This growth rate was much higher than the average growth rate of 11.2% from 2010 to 2022. Rapid economic growth may lead to increasing atmospheric BC concentrations. It is worth noting that in 2020, due to the impact of COVID-19, there was a significant decrease in human activities, so the BC concentration in 2020 was 17.20% lower than that in 2019.

The seasonal variation of BC concentration in Xinjiang was obvious (Fig. 3A). The atmospheric BC concentration in winter (145.52 ± 39.31 ng/m$^3$) was higher than that in autumn (124.95 ± 28.82 ng/m$^3$), spring (74.05 ± 9.96 ng/m$^3$), and summer (73.41 ± 5.69 ng/m$^3$). With the increase of temperature, the number of windy weather increases in spring (the average number of days in a month with a maximum wind speed of >5 m/s in spring was 24.4 days), which is beneficial for pollutant diffusion, so the BC concentration in spring and summer is low. In addition to the influence of wind speed, the wet deposition of summer precipitation and high vegetation coverage was significantly strengthened (the average monthly precipitation was 11 mm in summer and 5.9 mm in winter). Thus, the atmospheric BC concentration remained low in summer.

Figure 3B shows the monthly average BC concentration in Xinjiang. The BC concentration was the highest in December (292.99 ± 197.82 ng/m$^3$) and the lowest in June (135.69 ± 65.30 ng/m$^3$). The BC concentration showed an upward trend from August, reached the highest value in December, decreased rapidly from January to June, and increased slightly in July. The seasonal variation of BC concentration in Xinjiang was consistent with that of other regions of China, *i.e.,* Tianjin and Beijing (*Qin et al., 2019*; *Zhao et al., 2021*). The seasonal variation of BC emissions in certain regions is the main cause of the seasonal variation of atmospheric BC concentration in China (*Zhao, 2015*). The increase in the coal consumption for heating in winter, coupled with low wind
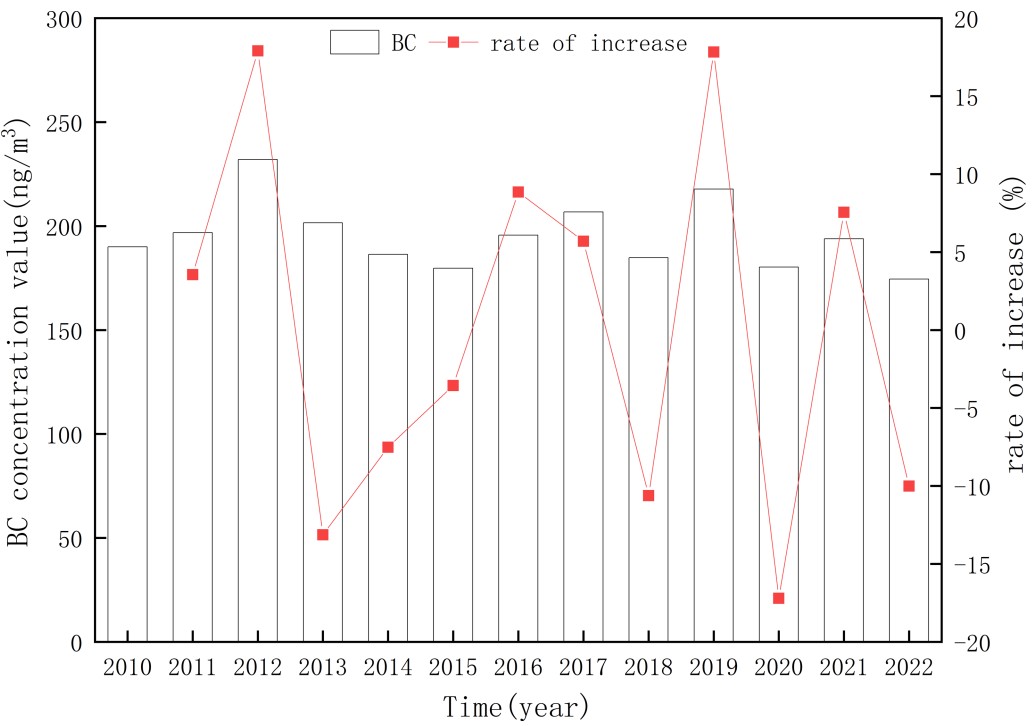

**Figure 2** Changes in the annual average and growth rate of atmospheric black carbon (BC) concentration in Xinjiang from 2010 to 2022.

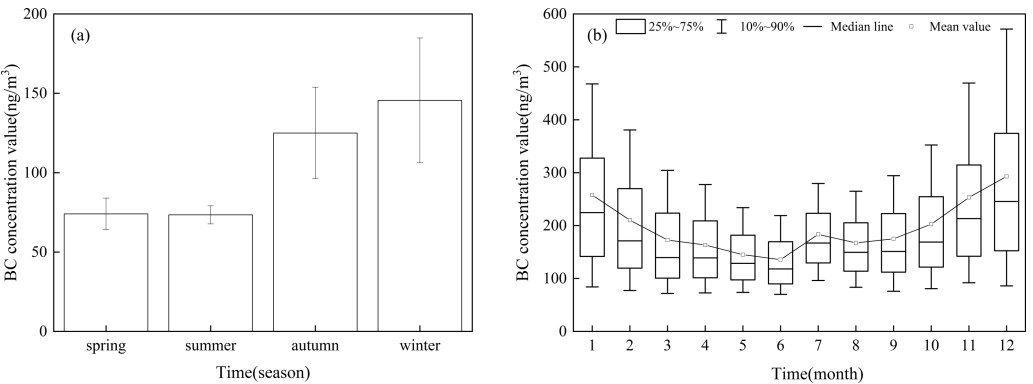

**Figure 3** Seasonal (A) and monthly (B) variation of atmospheric black carbon (BC) concentration in Xinjiang.

speed, frequent emergence of inversion layer, and low boundary layer height, inhibit the diffusion of BC and aggravate atmospheric BC pollution. In summer, the temperature rises, convective weather occurs frequently, and weather destabilization intensifies. Therefore, there are good diffusion conditions in both the vertical and horizontal directions, resulting in a decrease in BC concentration. However, in autumn, activities such as large-scale crop harvesting increase the BC concentration.

## Spatial distribution characteristics of atmospheric BC concentration

The analysis results of spatial distribution of annual average BC concentrations in Xinjiang from 2010 to 2022 using the MERRA-2 dataset (Fig. 4) showed that from 2010 to 2022, there were two areas with high BC concentrations: Urumqi-Changji-Shihezi urban agglomeration and Kashgar. The annual average BC concentration in Kashgar ($515.18 \pm 203.09$ ng/m$^3$) was higher than that in Urumqi-Changji-Shihezi urban agglomeration ($444.84 \pm 126.17$ ng/m$^3$), and much higher than that in Xinjiang ($196.53 \pm 57.90$ ng/m$^3$). The monthly average BC concentration in Kashgar was 2.09–2.77 times that of Xinjiang and 1.35–1.52 times that of Urumqi-Changji-Shihezi urban agglomeration (Fig. 5). The Urumqi-Changji-Shihezi urban agglomeration has developed industries and a dense population, with an urbanization rate of 86.98%. Although the Urumqi-Changji-Shihezi urban agglomeration only accounts for 5.7% of Xinjiang in area, it holds 83% of Xinjiang's heavy industries, its coal consumption accounts for 60% of Xinjiang's total, and the pollutant emissions account for one-third (Zhu et al., 2023). It was also found that in southern Xinjiang, the spatial distribution of BC concentration was highly consistent with the spatial distribution of industries and population. With Korla, Aksu, Kashgar and Hotan as important nodes, the high-BC-concentration areas surrounded the Tarim Basin. On the one hand, the special topographic and climatic conditions of the Tarim Basin leads to the formation of a unique layout of cities, and most of the human activities are concentrated in the cities, resulting in the highly consistency in distribution between high-BC-concentration areas and cities. On the other hand, the Tarim Basin is surrounded by mountains in the south, north, and west. The airflow from the east enters the Tarim Basin from the southern part of the Tianshan Mountains. After the airflow reaches the edge of the basin, it is not easy to spread due to the high mountains in the north, south, and west, resulting in the accumulation of pollutants in Aksu, Hotan, and Kashgar at the edge of the basin. Especially, Kashgar is located in the north of the westernmost Pamir Plateau, which is greatly affected by the frequent sand and dust in the Tarim Basin. Therefore, pollutants are easy to be transported to Kashgar, which is the main reason for its very high BC concentration.

From 2010 to 2022, the monthly average atmospheric BC concentration in Xinjiang showed significant seasonal differences (Fig. 6). The atmospheric BC concentration was high in autumn and winter and low in spring and summer. The area of high-BC-concentration areas (the northern slope of the Tianshan Mountains and the cities surrounding the Tarim Basin) was the largest in winter (November–February of the next year), and gradually decreased from November to February. The BC concentration also gradually decreased. The high-BC-concentration areas gradually decreased from March until it re-appeared in October. In spring and summer, all regions of Xinjiang have good pollutant diffusion conditions, with high wind speed and increased precipitation (the average number of days in a month with a maximum wind speed of >5 m/s in spring and summer was 23.5 days). It was worth noting that the seasonal atmospheric BC concentrations in the Urumqi-Changji-Shihezi urban agglomeration and Kashgar were higher than those in other regions of Xinjiang, and the monthly average concentrations were much higher than those of Xinjiang (Fig. 7). This indicates obvious spatial differences. Atmospheric pollution in the Urumqi-Changji-Shihezi urban agglomeration is more likely to occur in autumn

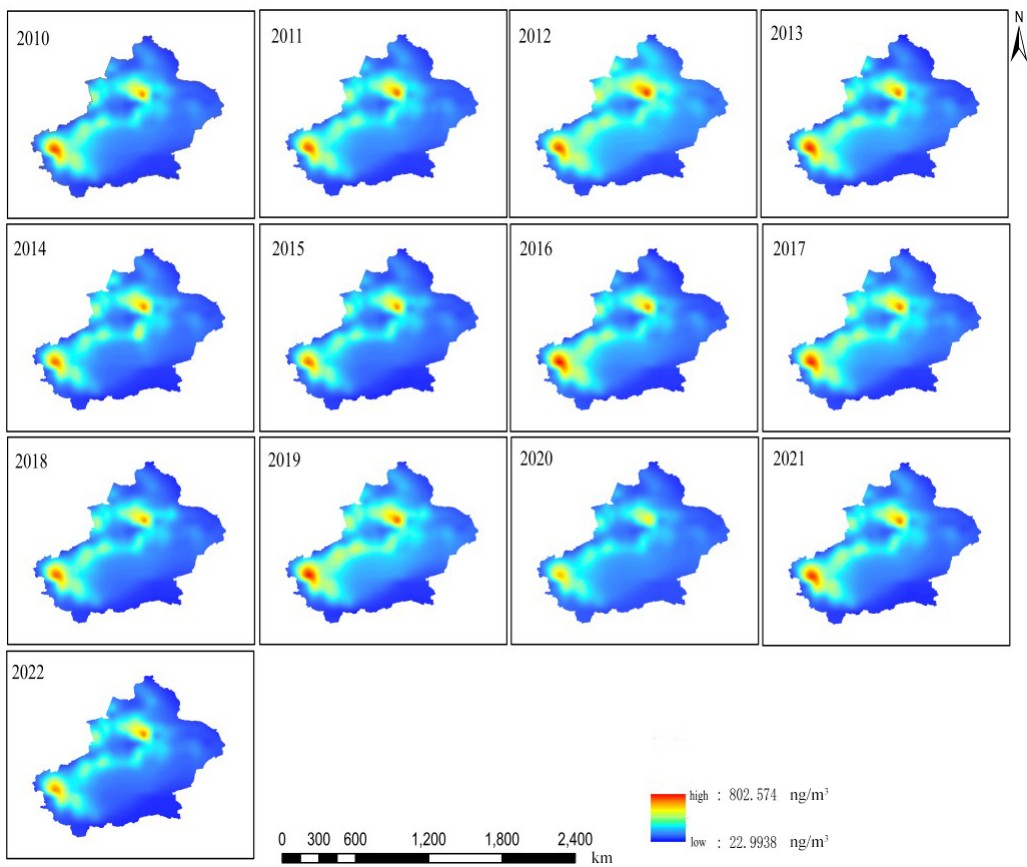

**Figure 4** Spatial distribution of annual average black carbon (BC) concentrations in Xinjiang from 2010 to 2022.

and winter (*Li, 2003*; *Ma et al., 2010*; *Wei, Meng & Li, 2012*). On the one hand, the high concentration of energy-intensive industries (with coal as the main energy) and human activities results in large emissions of pollutants. On the other hand, the heating season is long in winter. Due to the long-term impact of the Mongolian high, the weather is dry, the wind is slow, and the atmosphere is stable, which provide good conditions for the formation of an inversion layer. However, the inversion layer, coupled with the special topography, is not conducive to the diffusion of pollutants, (*Yang et al., 2011*; *Zhao, Li & Yang, 2011*; *Li et al., 2012a*; *Li et al., 2012b*), which makes atmospheric pollution more likely to occur in winter. Kashgar is located in the hinterland of the Eurasian continent, adjacent to the Taklamakan Desert and other deserts in Central Asia. $PM_{10}$ is the primary pollutant in Kashgar, and the average annual $PM_{10}$ concentration is 3.4 times the $PM_{10}$ concentration limit of China (24-hour average $PM_{10}$ concentration limit: 150 $\mu g/m^3$). Due to there is a lot of sand and dust in spring and summer, dust particles may transport BC to Kashgar, leading to a high BC concentration.

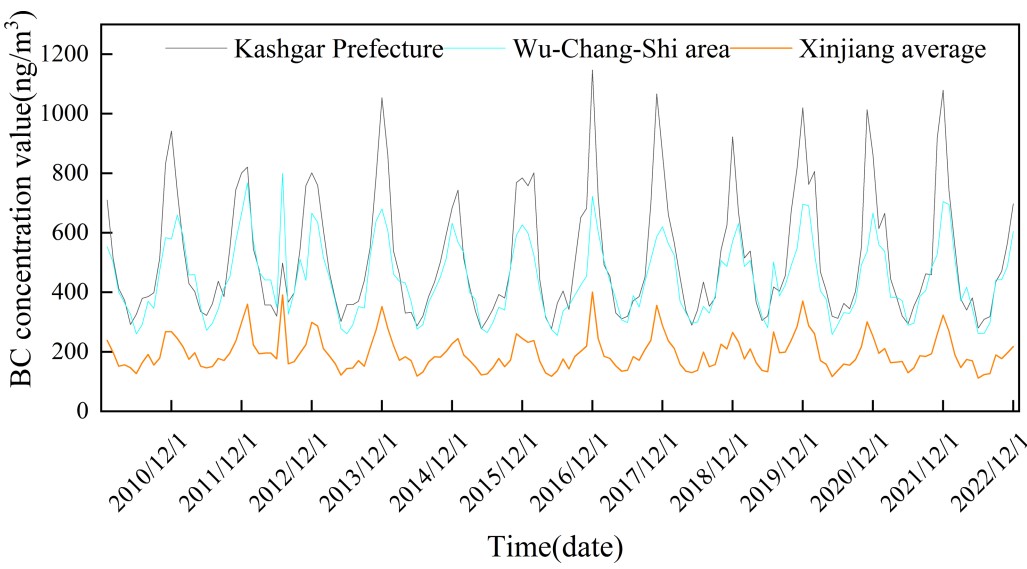

**Figure 5** Changes of monthly average black carbon (BC) concentration in Kashgar, Urumqi-Changji-Shihezi urban agglomeration (Wu-Chang-Shi area), and Xinjiang.

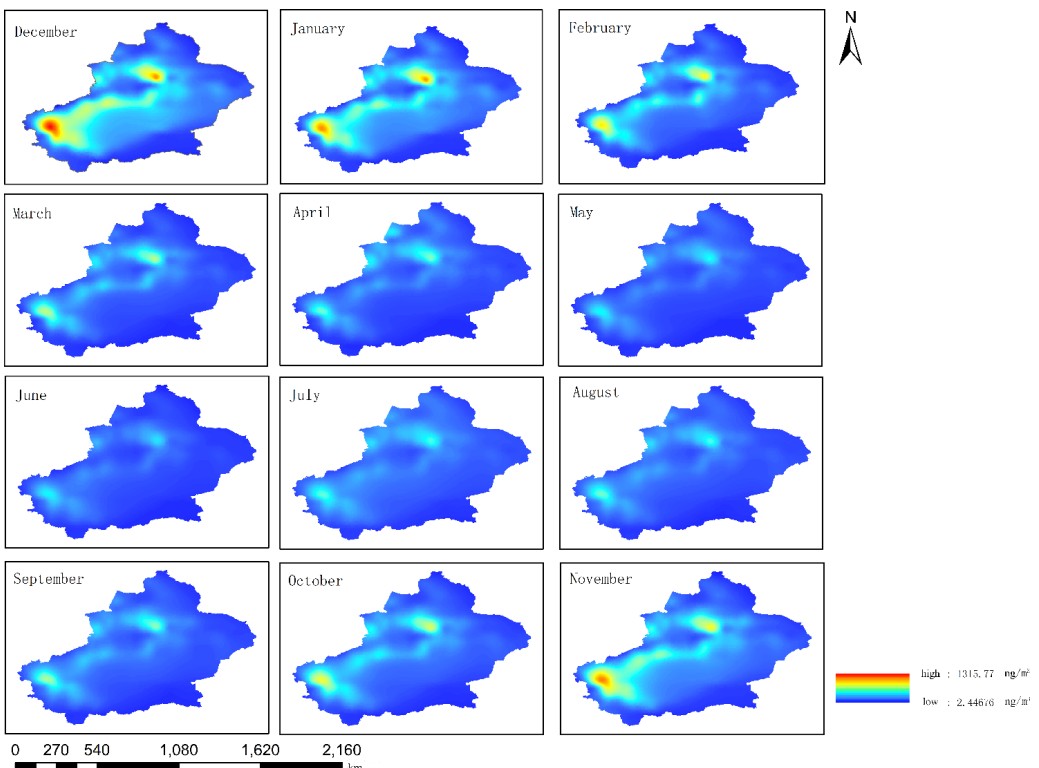

**Figure 6** Distribution of the monthly average atmospheric black carbon (BC) concentrations from 2010 to 2022 in Xinjiang.

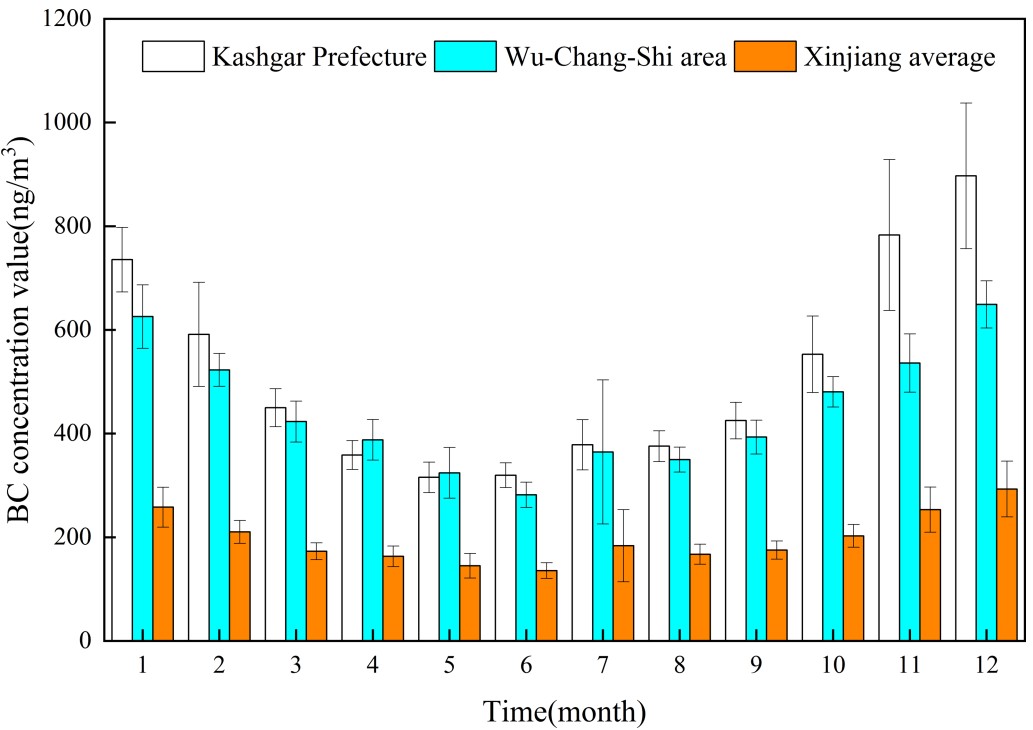

**Figure 7  Comparison of monthly average black carbon (BC) concentrations in Kashgar, Urumqi-Changji-Shihezi urban agglomeration, and Xinjiang.**

## Influences of meteorological and environmental factors on atmospheric BC concentration

*Jeong & Park (2013)* showed that meteorological conditions caused particulate matter concentrations to decrease by 4% in summer but increase by 7% in winter in East Asia. The results of the RF model predicting the influence of meteorological factors on the atmospheric BC concentration (Fig. 8) showed that the correlation coefficient between the atmospheric BC concentration and the simulated value was 0.76, and the importance of meteorological factors to the atmospheric BC concentration was ranked as follows: snow depth >land surface temperature >humidity >sunshine hours >wind speed. Therefore, the atmospheric BC concentration at the Akdala Station was affected by all meteorological factors.

The snow depth was 0–24.04 cm at the Akdala station (average: 11.33 cm), which was divided into six grades (Fig. 9A), to explore the variation of atmospheric BC concentration with snow depth. It was found that the atmospheric BC concentration decreased with the increase of snow depth, and the average BC concentration in the snow cover season was 169.62 ng/m³, which was much lower than that in the snow cover-free season. The number of days without snow cover accounted for 71.78% of the total, and the number of days with
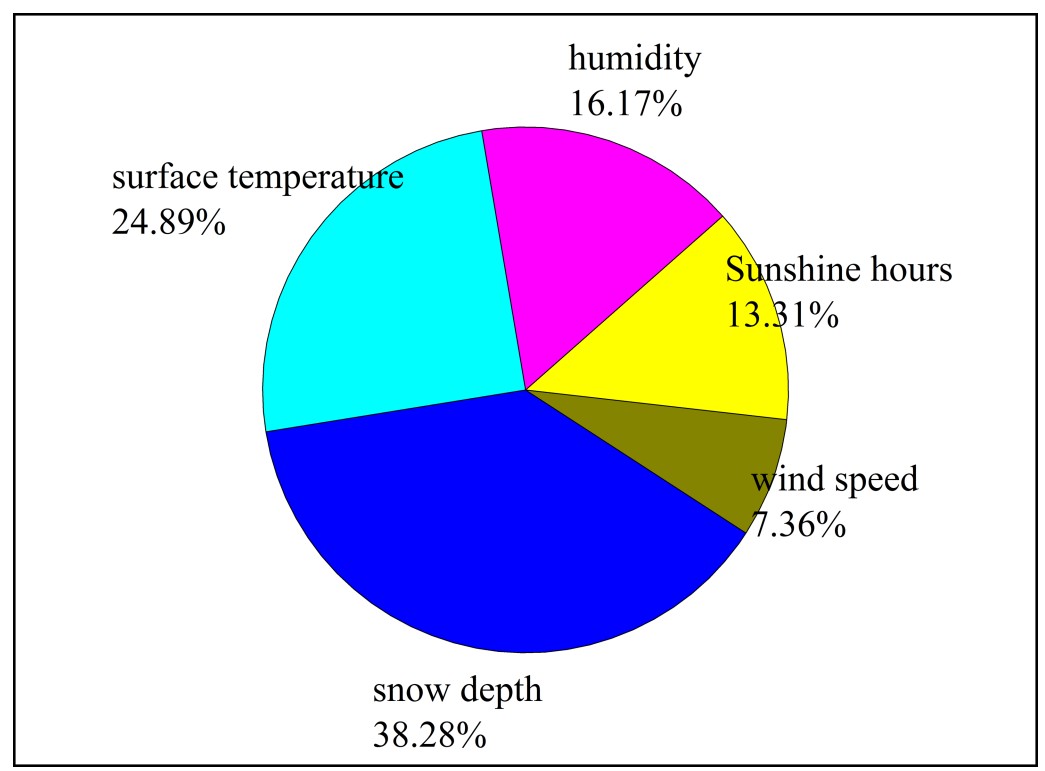

**Figure 8**  Ranking of the importance of meteorological factors to atmospheric black carbon (BC) concentration.

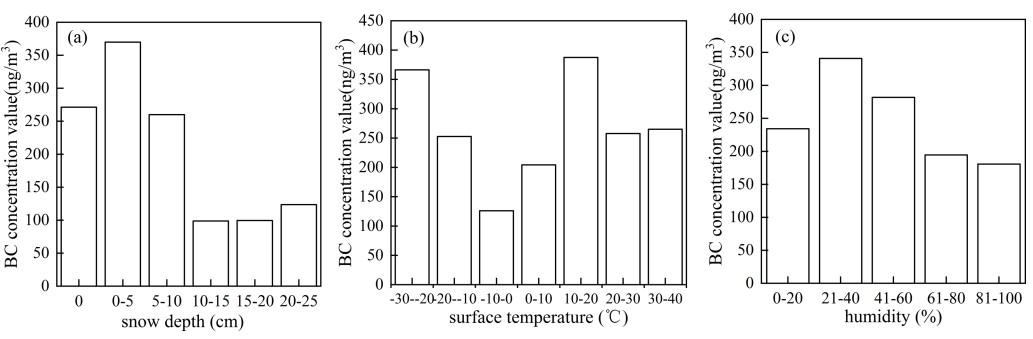

**Figure 9**  Variations of atmospheric black carbon (BC) concentration with snow depth (A), land surface temperature (B), and humidity (C).

0–5, 5–10, 10–15, 15–20, and 20–25 cm snow depth accounted for 4.93%, 1.64%, 11.51%, 6.03%, and 1.37%, respectively.

The land surface temperature ranged from −22.81 to 38.19 °C (average: 10.18 °C), which was divided into 7 grades (Fig. 9B), to explore the variation of atmospheric BC concentration with land surface temperature. The number of days with land surface temperature at −30 to −20 °C, −20 to −10 °C, −10–0 °C, 0–10 °C, 10–20 °C, 20–30 °C,

and 30–40 °C accounted for 1.09%, 11.51%, 20.82%, 16.71%, 14.52%, 20.55%, and 14.79%, respectively. Taking 0 °C as the cut-off point, when the land surface temperature was lower than 0 °C, the atmospheric BC concentration decreased with the increase of land surface temperature; when the land surface temperature was above 0 °C, the BC concentration increased with the increase of land surface temperature. The atmospheric BC concentration was the highest at 10–20 °C. The land surface temperature was below 0 °C in winter. Most areas of Xinjiang are located in the rear of the Mongolian high, with stable stratification and slow winds, which inhibits the diffusion of pollutants. Besides, due to the blocking by the Tianshan Mountains, there is a temperature inversion in northern Xinjiang, which promotes the accumulation of pollutants. In winter, the demand for heating increases, industrial coal-fired boilers and automobile exhausts emit large amounts of pollutants, causing an increase in the concentration of BC.

The relative humidity ranged from 15.63% to 92.83% (average: 60.08%), which was divided into 5 grades (Fig. 9C), to explore the variation of atmospheric BC concentration with humidity. The atmospheric BC concentration decreased with the increase of humidity in a certain range. The days with humidity at 0%–20%, 20%–40%, 40%–60%, 60%–80%, and 80%–100% accounted for 1.09%, 12.88%, 35.07%, 38.08%, and 12.88%, respectively. High relative humidity causes aerosols in the atmosphere to absorb moisture and settle, which reduces the atmospheric BC concentration. Currently, climate of Northwest China tends to be warmer and wetter due to the influence of climate warming, which accelerates the deposition and transformation of air pollutants, causing a decreasing in the annual variation of atmospheric BC concentration.

The results of the RF model predicting the influence of environmental factors on the atmospheric BC concentration (Fig. 10) showed that the correlation coefficient between atmospheric BC concentration and simulated value was 0.71, and the importance of environmental factors to atmospheric BC concentration was ranked as follows: $NO_2$ >$PM_{10}$ >$SO_2$ >CO >$PM_{2.5}$ >$O_3$.

The $NO_2$ concentration at Akdala Station ranged from 0.02 to 4.26 ppb (average: 1.84 ppb), which was divided into five grades (Fig. 11A), to explore the variation of atmospheric BC concentration with $NO_2$ concentration. It was found that the atmospheric BC concentration increased with the increase of $NO_2$ concentration. The days with $NO_2$ concentration of 0–1 ppb, 1–2 ppb, 2–3 ppb, 3–4 ppb, and 4–5 ppb accounted for 10.69%, 49.66%, 34.83%, 3.45%, and 1.38%, respectively.

The $PM_{10}$ concentration at Akdala Station ranged from 1.3 to 31.8 μg/m³ (average: 17.71 μg/m³), which was divided into 7 grades (Fig. 11B), to explore the variation of atmospheric BC concentration with $PM_{10}$ concentration. It was found that the atmospheric BC concentration increased with the increase of $PM_{10}$ concentration. The number of days with $PM_{10}$ concentration of 0–5 μg/m³, 5–10 μg/m³, 10–15 μg/m³, 15–20 μg/m³, 20–25 μg/m³, 25–30 μg/m³, and 30–35 μg/m³ accounted for 1.94%, 10.28%, 20.28%, 29.72%, 25.83%, 9.72%, and 1.94%, respectively.

The $SO_2$ concentration at Akdala Station ranged from 0.43 to 2.33 ppb (average: 1.38 ppb), which was divided into four grades (Fig. 11C), to explore the variation of atmospheric BC concentration with $SO_2$ concentration. It was found that the atmospheric

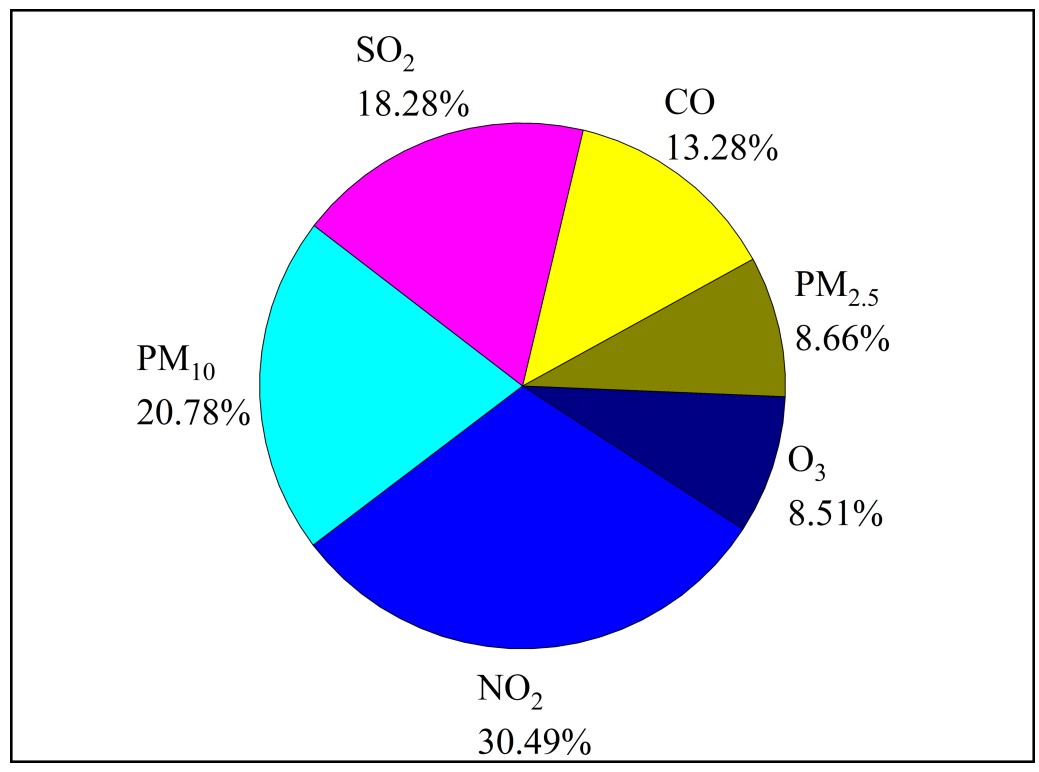

**Figure 10  Ranking of the importance of environmental factors on atmospheric black carbon (BC) concentration.**

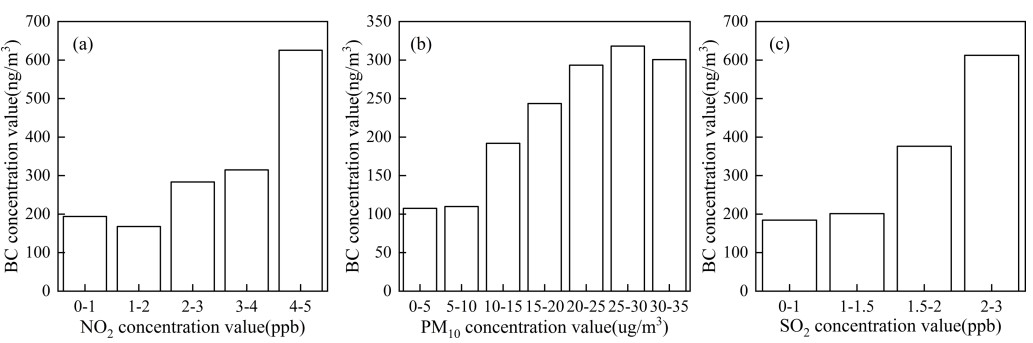

**Figure 11  Variations of atmospheric BC concentration with $NO_2$ (A), $PM_{10}$ (B), and $SO_2$ (C) concentrations.**

BC concentration increased with the increase of $SO_2$ concentration. The number of days with $SO_2$ concentration of 0–1 ppb, 1–1.5 ppb, 1.5–2 ppb, and 2–3 ppb accounted for 0.86%, 74.43%, 24.14%, and 0.57%, respectively.

Black carbon is a typical primary aerosol that is emitted directly from natural and anthropogenic sources. Factors such as local pollutant sources, emission intensity, meteorological conditions, and geographical environment lead to strong spatiotemporal

variation of BC pollution. In terms of meteorological conditions, the unique arid climate and geographical characteristics of Xinjiang is conducive to the diffusion and wet deposition of BC in a certain temperature range ($-10$–$10$ °C) and high humidity conditions, and the BC concentration may be high when the depth of the snow cover is 0–10 cm. The concentrations of PM, $SO_2$, CO, and $O_3$ in the atmosphere are the results of anthropogenic emissions. Through the analysis of the influence of major air pollutants on atmospheric BC concentrations, it was found that $NO_2$, $PM_{10}$, and $SO_2$, which ranked in the top three by importance, were all products of industrial and anthropogenic emissions, and the atmospheric BC concentration increased with the increase of $NO_2$, $PM_{10}$, and $SO_2$ concentrations.

## CONCLUSION

(1) The reliability of the MERRA-2 reanalysis data in exploring the spatiotemporal variation characteristics of atmospheric BC concentration in Xinjiang was high (RAD = 0.65).

(2) From 2010 to 2022, the annual average atmospheric BC concentration in Xinjiang was $195.40 \pm 15.55$ ng/m$^3$, with a multi-year average change rate of $-0.05\%$. The highest value (231.99 ng/m$^3$) was appeared in 2012, and the lowest value (174.50 ng/m$^3$) was appeared in 2022. In 2020, due to the impact of the COVID-19, there was a significant decrease in human activities, so the atmospheric BC concentration in 2022 was 17.20% lower than that in 2021.

(3) The atmospheric BC concentrations in Xinjiang in winter ($145.52 \pm 39.31$ ng/m$^3$) was higher than that in autumn ($124.95 \pm 28.82$ ng/m$^3$), spring ($74.05 \pm 9.96$ ng/m$^3$), and summer ($73.41 \pm 5.69$ ng/m$^3$). The atmospheric BC concentration showed an increasing trend from August, reached the highest value in December, decreased rapidly from January to June, and then increased slightly in July.

(4) The annual and monthly average atmospheric BC concentrations showed obvious spatial differences from 2010 to 2022 in Xinjiang. There were two high-BC-concentration areas: The northern slope of the Tianshan Mountains and the cities surrounding the Tarim Basin. Especially, the atmospheric BC concentration in Urumqi-Changji-Shihezi urban agglomeration and Kashgar were very high.

(5) The changes of atmospheric BC concentration were related to meteorological and environmental factors. Snow depth, land surface temperature, and humidity were the main meteorological factors affecting atmospheric BC concentration, and $NO_2$, $PM_{10}$, and $SO_2$ were the main environmental factors affecting atmospheric BC concentration. The RF model and the comprehensive analysis of the spatiotemporal variation characteristics of atmospheric BC concentration showed that atmospheric BC concentration was greatly affected by human activities, climate change, and other factors.

### Funding

This work was financially supported by the Desert Weather Scientific Research Fund of China (Sqj2022021) and Major scientific and technological projects of the XPCC: Research

and application demonstration of key technologies of heavily polluted weather cause, influence mechanism analysis, detection and early warning, and comprehensive treatment in major cities on the north slope of the Tianshan Mountains (2023AB036). The funders had no role in study design, data collection and analysis, decision to publish, or preparation of the manuscript.

## Grant Disclosures

The following grant information was disclosed by the authors:

The Desert Weather Scientific Research Fund of China: Sqj2022021.

Major scientific and technological projects of the XPCC: Research and application demonstration of key technologies of heavily polluted weather cause, influence mechanism analysis, detection and early warning, and comprehensive treatment in major cities on the north slope of the Tianshan Mountains: 2023AB036.

## Competing Interests

The authors declare there are no competing interests.

## Author Contributions

- Xiang Xie conceived and designed the experiments, performed the experiments, analyzed the data, prepared figures and/or tables, authored or reviewed drafts of the article, and approved the final draft.
- Zhujun Zhao conceived and designed the experiments, performed the experiments, analyzed the data, prepared figures and/or tables, authored or reviewed drafts of the article, and approved the final draft.
- Shuting Li analyzed the data, prepared figures and/or tables, and approved the final draft.
- Zhongqi Lu performed the experiments, prepared figures and/or tables, and approved the final draft.
- Lijuan Chen analyzed the data, authored or reviewed drafts of the article, and approved the final draft.
- Jiangjiang Cai analyzed the data, authored or reviewed drafts of the article, and approved the final draft.

## Data Availability

The BC concentration monthly data (MERRA-2) is available at NASA Earth Science Data: https://disc.gsfc.nasa.gov/datasets/M2TMNXAER_5.12.4/summary?keywords=MERRA2

DOI: https://doi.org/10.5067/FH9A0MLJPC7N

The scripts for the BC datasets (M2T1NXAER and M2tmnxaer) are available at NASA and the hourly BC concentration data was downloaded in bulk from the command-line program (Cygwin).

## Supplemental Information

Supplemental information for this article can be found online at http://dx.doi.org/10.7717/peerj.18187#supplemental-information.

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
