# Peer review of "Spatiotemporal variations of atmospheric black carbon concentration and its correlation with meteorological and environmental factors in Xinjiang, China, from 2010 to 2022"

_PeerJ, doi:10.7717/peerj.18187_

## Round 0.1 · original submission · Major Revisions

Atmospheric black carbon (BC) is one of the hottest topics in the field of atmospheric science, impacting climate change, atmospheric environment and human health. I consider this study as a significant contribution to the topic of BC in northwest China.

Please revise this manuscript carefully based on these comments. The academic contribution and discussion section should be improved in the revision.

Reviewer 1 ·

Basic reporting

This manuscript well written in structure, but still needs some improvements in language expressions and literature references.

Experimental design

This manuscript investigates the temporal and spatial variations of atmospheric black carbon (BC) concentration and its correlation with meteorological and environmental elements in Xinjiang from 2010 to 2022. The Mann-Kendall (MK) test and random forest (RF) model are used with MERRA-2 reanalysis data and ground-based observation data. It is found that, the annual average concentration of atmospheric BC in Xinjiang was 195.40 ± 15.55 ng/m3, with a multi-year average change rate of -0.05%. The BC concentration also showed significant seasonal diversity and obvious spatial variations. Besides, meteorological factors such as snow depth, surface temperature, and humidity, along with environmental factors like NO2, PM10, and SO2, were found to influence the BC concentration changes.

Validity of the findings

Atmospheric BC is one of the hottest topics in the field of atmospheric science, impacting climate change, atmospheric environment and human health. I consider this study as a significant contribution to the topic of BC in northwest China.

Annotated reviews are not available for download in order to protect the identity of reviewers who chose to remain anonymous.

Reviewer 2 ·

Basic reporting

Line 41: It is not correct expression. Please make sure the use of the carbonaceous matter.
Line 52: please verify whether “heterogeneous reactions” can be used here.

Experimental design

no comment

Validity of the findings

Line 192: This proves that the changes of BC concentration are positively correlated with human activities, especially industrial activities. It is not a collect conclusion. If the relationship between human or industrial activities and BC concentration is so easy and right, please author explain why PM2.5 concentration in Wuchangshi is comparable with Beijing-Tianjin-Hebei region and northern Henan.

Line 194, please legitimately explain the reason causing a decreasing trend of BC concentration in the past 13 years. Line 201-203 is opposite to Line 194. You should know that the GDP of Xinjiang is continuing to increase in the past 13 years and Xinjiangís industry still relies greatly on fossil fuels, which means that the BC concentration in the past 13 years should increase according to the logic from the author.

Line 215-216 please provide the data to support that the number of windy weather increases in spring

Line 217 please provide the data of the average humidity in winter and the precipitation in summer and winter.

Line 222-223, please state the specific cities and add the reference.

Line 243, can not understand what is “theis may”

Line 275-276 has the same problem as Line 215-216

Line 307-308: you find that snow depth is the most important meteorological factor for influencing the BC concentration. You should know that Xinjiang has the deepest snow depth in winter while the BC concentration in winter is highest. Additionally. Please author make the content to be consistent. You first contribute the low BC concentration in summer to the windy weather and humidity while you then rank the wind speed to the last one. It is a little unbelievable.

Reviewer 3 ·

Basic reporting

Authors have used the MERRA-2 based BC datasets to study the temporal and spatial variation in Xinjiang province of China. This is praiseworthy considering the vastness of the area and lower number of studies based in that province. Nevertheless, current version of the manuscript needs a thorough edit on the language use. Another major concern is the lack of scientific discussions. Authors have well-presented their findings but the discussion of their result is missing.
Other suggestions have been listed here:
- L25: in China?
- L74-77: Provide references
- L79: atmospheric, please correct.
- L86: And? Not a good way to start a sentence.
- Introduction: More literature review is needed and should be updated with (at least) national studies based on MERRA-2 BC analysis.
- L102: reads incomplete.
- Section 2.3: Met data and other air quality data were used only for a year? 2022?
- I would suggest to include a map of country showing the study area and monitoring site.
- If there are any abbreviations used in the figure, please explain it in the caption. Also explain if you have different names for legends and in caption (sucha s in Figure 5).
- L243: Theis?
- L246: 1/3 should be replaced by one-third.

Experimental design

Please see the comments in Section 1.

Validity of the findings

Please see the comments in Section 1.

Additional comments

Please see the comments in Section 1.

---

## Round 0.2 · accepted · Accept

It is much better than before and I am happy to Accept this submission.

Reviewer 1 ·

Basic reporting

Comments for the manuscript “Spatiotemporal variations of atmospheric black carbon concentration and its correlation with meteorological and environmental factors in Xinjiang, China, from 2010 to 2022 (#97084)”:

This manuscript investigates the tempo-spatial variations of atmospheric black carbon (BC) concentration and its correlation with meteorological and environmental elements in Xinjiang from 2010 to 2022. The Mann-Kendall (MK) test and random forest (RF) model are used with MERRA-2 reanalysis data and ground-based observation data. It is found that, the annual average concentration of atmospheric BC in Xinjiang was 195.40 ± 15.55 ng/m3, with a multi-year average change rate of -0.05%. The BC concentration also showed significant seasonal diversity and obvious spatial variations. Besides, meteorological factors such as snow depth, surface temperature, and humidity, along with environmental factors like NO2, PM10, and SO2, were found to influence the BC concentration changes.

This manuscript is the revised version, and all the revision suggestions I put forward have been well responded and revised. Therefore, I believe that this manuscript deserves consideration for acceptance.

Experimental design

The experimental design and research questions of this manuscript are well defined, relevant and meaningful. The investigation of this manuscript has been conducted rigorously.

Validity of the findings

Atmospheric BC is one of the hottest topics in the field of atmospheric science, impacting climate change, atmospheric environment and human health. I consider this study as a significant contribution to the topic of BC in northwest China.